# Investigating the Absorption Spectra of a Plasmonic Metamaterial Absorber Based on Disc-in-Hole Nanometallic Structure

**DOI:** 10.3390/nano12203627

**Published:** 2022-10-16

**Authors:** Amr M. Mahros, Yara Alharbi

**Affiliations:** 1Department of Engineering Physics, Alexandria University, Alexandria 21544, Egypt; 2Department of Physics, University of Jeddah, Jeddah 21432, Saudi Arabia

**Keywords:** plasmonics, metamaterial absorber, FDTD, optical absorption

## Abstract

In this work, we present and explore the characteristics of a plasmonic metamaterial absorber based on a metal–insulator–metal functional stack. The proposed structure consists of glass “sandwiched” between a silver reflector and a titanium metallic disc, embedded inside a Ti periodic nano-hole array, as an outside layer. In the visible and infrared regimes, the optical absorption spectra of such structures have been investigated using the finite difference time domain method. The impact of modifying nano-hole and embedded disc diameters on the absorber’s performance has been investigated. Changing these two distinct structural parameters tunes the coupling effect between the localized and propagating surface plasmons. The adequate bandwidth, average spectral absorption rate, and short circuit current density are calculated to determine the performance of the designated absorber. The proposed structure of the plasmonic metamaterial absorber reaches an average absorption of over 94% in a bandwidth of 0.81 µm and near-perfect absorption of 98% around the wavelength of 0.7 µm, with an almost 100% relative absorption bandwidth and 41 mA/cm^2^ short circuit current density. In addition, the results show that the disc-in-hole absorber’s structural parameters can be changed precisely and facilely to tailor to the absorption spectra.

## 1. Introduction

Over the last decade, renewable energy resources such as solar, hydropower, and wind have been used in defeating environmental pollution and energy shortages. In this regard, solar cells have received attention as a solution to environmental pollution and energy shortage problems. Solar cells produce energy by converting sunlight directly into electrical energy. However, the main disadvantage of using solar cells is poor light-absorption efficiency. Therefore, increasing light trapping plays an essential role in enhancing the efficiency of solar cells by increasing their optical path length. There are several light-trapping techniques introduced in previous works, such as antireflecting coatings [1,2], photonic crystals [3,4], surface texturing [5], and metallic nanoparticles [6,7,8]. These techniques reduce reflection on the front surface and improve the optical path length.

Recently, plasmonic metamaterials have drawn attention due to their unique absorption properties [9,10,11]. Incorporating plasmonic metamaterials into solar cells can resolve low absorption efficiency and short circuit current problems. The plasmonic effect is the interaction of incident light and the free electrons of a metallic nanostructure. This effect is appropriate for applications such as solar cells, sensing, infrared spectroscopy, LEDs, and thermal imaging [12,13]. In this regard, plasmonic nanostructures can be integrated on the front surface of the cell because, in this case, they are reducing the reflections from the surface and increasing the optical path lengths by scattering light into the active layer due to the localized surface plasmon (LSPs) [14]. The performance of solar cells is affected by the nanostructure designs that have been used, such as nano-holes [15,16], nanorods [17], nanopyramids [18], and nanowires [19]. Furthermore, previous research found that localized surface plasmons (LSPs) are sensitive to the type of material, such as silver [20,21], gold [6,22], indium [23], and aluminum [7]. In addition, localized and propagating surface plasmons are highly affected by the size [7], periodicity [7], and shape [24] of the nanostructure.

The implication of plasmonic nanostructures inside semiconductors has been explored as it causes an increase in the efficiency of solar cells [25]. Silver and gold nanoparticles attracted researchers’ attention in relation to sunlight harvesting by localized surface plasmons. Their plasmonic resonance is in the visible region. Al and Ti have favorable compatibility and unique scattering support with plasmonic resonances on the ultraviolet and near-infrared spectra [26]. Since most metamaterials are made from metals, the plasmonic effect of metals is extremely crucial in metamaterial absorbers [27]. Metamaterial absorbers are manufactured in multilayer, two-layer, three-layer [26], and four-layer structures to enhance their optical absorption [28,29]. Three-layer metamaterial absorbers are the most commonly used variant [28]. They are composed of metal–insulator–metal stacks. The top layer of the metamaterial absorber is a metal in a periodic pattern followed by a dielectric layer, and the bottom layer is a metal reflector [29,30]. The thickness of the third layer is crucial in preventing light transmission.

In this paper, we designed a plasmonic metamaterial absorber based on metal–insulator–metal structure of titanium (Ti)–glass (BK_7_)–silver (Ag) to analyze the enhancement of the optical absorption in the visible and infrared regions. The simulation was carried out with OptiFDTD 16 RC simulation tool from Optiwave Inc. We investigated the effects on absorption using various metallic nano-disc sizes and different nano-hole sizes. Then, we identified the impact on absorption caused by merging a metallic nano-disc in a hole by considering the effect of changing the diameters of the Ti disc and the hole.

## 2. Methodology

Figure 1 presents the schematic unit cell of the proposed plasmonic metamaterial absorber consisting of three functional layers. A hollow cylindrical metal–insulator cap is mounted on a metallic back reflector substrate. The 250 nm bottom silver mirror has a pitch of W = 250 nm. The titanium–glass cap has an inner diameter of d_2_ = 120 nm and an outer diameter of d_3_ = 240 nm. The titanium plasmonic layer has a thickness of 20 nm, and the BK7 spacer has a thickness of 80 nm. Finally, a Ti disc-in-hole of diameter d_1_ = 80 nm is positioned at the center of the cylindrical cap. The refractive index of the dielectric spacer was obtained from the literature from the OptiFDTD material library [31]. The relative permittivity *εr* (*ω*) of the dispersive metallic top layer and back reflector substrate was determined using the Lorentz–Drude model [32,33]:(1)$$\varepsilon_{r}\left(\omega \right)=\varepsilon_{\infty }+\overset{N}{\underset{m=1}{\sum }}\frac{f_{m}\omega_{om}^{2}}{\omega_{om}^{2}-\omega^{2}+i\omega \Gamma_{m}}$$
where *ε_∞_* denotes the permittivity at infinite frequency, *f_m_* is a function of position specifying the oscillator strengths, and *Γ_m_* is the damping coefficient. The incident wave frequency and the resonant frequencies are represented by *ω* and *ω_om,_* respectively.

The simulation was carried out by solving Maxwell’s curl equations of different materials using the finite difference time domain algorithm. The FDTD method was performed using the electromagnetic solver OptiFDTD simulation tool from Optiwave Inc.

Arrays of metal−insulator−metal (MIM) multilayer structures can support more affluent optical properties because of the interplay between SP modes at different metal−insulator interfaces. Surface plasmon resonance could be excited by the incident light on the metallic nano-layer. The thin disk dielectric resonator accomplishes electric and magnetic dipole resonances [34].

Adding an ultrathin dielectric coating to a reflective substrate enabled perfect light absorption. Light is entirely trapped due to amplitude splitting destructive interference. As the top and bottom metal layers are highly reflective, asymmetric Fabry–Perot conditions can be met. The maximum absorption resonance wavelength for the insulator–metal stack is calculated by [13]:(2)$$\lambda_{max}=2\pi n_{i}t\tan\left(n_{i}\sqrt{\frac{n_{m}-n_{0}}{n_{0}\left(n_{i}{}^{2}-n_{m}n_{0}\right)}}\right)\;,$$
where $n_{i}$ is the insulator refractive index, t represents the thickness of the dielectric material, $n_{m}$ represents the substrate refractive index, and $n_{0}$ represents the superstrate refractive index. In addition, the absorption, A, is determined by using the following equation:(3)A=1−T−R,
where T is the transmission, and R is the reflection.

We used a linearly polarized Gaussian modulated plane wave source with a Bloch signal of 500 THz center frequency and 375 THz full width half maximum to realize a broadband simulation in visible and near-infrared regions. The light pulse in the time domain has an offset time of 4.2 × 10^−15^ s and a half-width of 2.3 × 10^−15^ s. We have established a simulation wafer in Cartesian coordinates x, y, and z with periodic boundary conditions (PBC) in the x- and y-directions to avail the periodicity of the MA unit cell. An anisotropic perfect matching layer (PML) was used in the z-direction to serve as an absorbing boundary condition. The simulation was performed at normal incidence. Two x–y observation areas were used, 100 nm behind the source and 50 nm after the structure, to calculate the reflectance (R) and transmittance (T) spectra through the designed absorber. One can employ the electric field distribution recognized in the simulation waver to calculate the absorption per unit volume in each cell using Equation (4). Then, the absorbed power is normalized by the incident source power [5]:(4)$$P_{abs}=-0.5\;Re\;\left(\mathrm{div}\;S\right)=-0.5\;\omega \;{\left|E\right|}^{2}\;Im\;\left(\varepsilon \right)$$
where **S** denotes the Poynting vector, |**E**| is the magnitude of the electric field intensity on the selected monitor, and *ε* specifies the material dielectric constant. The incident wave frequency is represented by *ω*.

## 3. Results and Discussion

In this section, we first examine the absorption spectra of the reported plasmonic metamaterial absorber throughout the wavelength window (0.3 µm–3 µm) in two featured cases when the metal–insulator cap is a nano-hole or nano-disc array.

The top view of the nano-disc array (d_2_ = d_3_) of the titanium–glass cap is shown in Figure 2a. The modulation of the absorption spectra due to the variation of the nano-disc diameter d_3_ are illustrated in Figure 2b,c presents the absorption spectra realized by three nano-disc arrays of various diameters (d_1_ = 30 nm, 120 nm, and 220 nm).

The absorption spectra exhibit an enhanced peak that indicates less reflected optical energy due to perfect matching between the plasmonic metal–insulator–metal structure and free space. This crest may be associated with metallic localized plasmon resonance [10]. As the nano-disc diameter, d_1_, increases, one can notice the broadening of the absorption peak. The titanium top layer is highly lossy, which furnishes a low-quality factor that causes broadening. The avoided resonance crossing dip is not a zero because the coupling efficiency is not strong enough. As can be seen, the absorption spectra exhibit a pronounced avoided resonance crossing dip, not a zero, at almost 0.375 µm because the coupling efficiency between plasmonic resonance and electric/magnetic field is not that strong. These results harmonize with the plasmonic surface lattice resonances, which occur at a wavelength slightly longer than the diffraction edge of nP for normal incidences, where n is the refractive index of glass [10].

Figure 3a displays the top view of the titanium–glass cap nano-hole array (d_1_ = 0), while the modification of the absorption spectra due to the change in the nano-hole diameter d_2_ is presented in Figure 3b,c showing the absorption spectra recognized by three nano-hole arrays of different diameters (d_2_ = 30 nm, 120 nm, and 220 nm).

The absorption spectra exhibit the pronounced avoided resonance crossing dip at 0.375 µm. Figure 3 presents an absorption peak for the red trace (d_2_ = 120 nm) at 0.6 µm. Plasmon resonance occurs through multiple reflections at the inside walls due to Fabry–Perot excitations. As the nano-hole diameter increases, the absorption peak blue shifts. For the blue trace (d_2_ = 220 nm), the absorption crest is at 0.5 µm because the distance between the Fabry–Perot reflections increases. In rapprochement to the case of using nano-disc patterns, the outcomes of Figure 3 display the existence of a broadening peak that may be associated with propagating plasmon resonance [34,35]. However, in sharp contrast to the case of nano-disc arrays, the broadening becomes irritated as the nano-hole diameter d_2_ decreases. In this investigation, we capitalize on the antagonistic behavior of the nano-disc and hole arrays to tailor the absorption spectra.

For absorbance over 85%, the effective bandwidth *BW* is calculated as the difference between the upper wavelength, *λ_U_*, and the lower wavelength, *λ_L_*. Then, to examine the absorption efficiency and performance, the relative absorption bandwidth (*RAB*) and the average spectral absorption rate (*SAR*) are calculated numerically, in that regime, using Equations (5) and (6) [36,37]:(5)$$RAB=2\frac{BW}{\lambda_{U}+\lambda_{L}}$$
(6)$$SAR=\frac{\int_{\lambda_{L}}^{\lambda_{U}}A\left(\lambda \right)d\lambda }{BW}$$

Figure 4 presents the absorption properties of plasmonic metamaterial absorbers, designed with two featured shapes of metal-insulator cap, under normal incidence.

Figure 4a,b show that bandwidth and relative absorption bandwidth decrease as the nano-disc (d_2_ = d_3_) diameter increases. However, they increase as the nano-hole (d_1_ = 0) diameter increases, as illustrated in Figure 4d,e. Figure 4e,f present that the plasmonic metamaterial absorber based on nano-disc periodic structure exhibits a higher average spectral absorption rate than those based on the nano-hole array structure. Figure 4c,f presents the average spectral absorption rate collected in nano-disk and nano-hole patterns, respectively. As the nano-disc diameter d_1_ increases, one can notice the broadening and growing of the absorption peak. On the other hand, as the nano-hole diameter, d_2_, increases, contracting and lessening of the absorption crest is observed. This behavior resulted in a dip in the average spectral absorption rate falling close to the middle of the diameter range (125 nm).

Secondly, we looked at the absorption performance of the designed metamaterial disc-in-hole nano-metallic structure. Modifying the inner nano-disc diameter and outer nano-hole diameter may tune the out-of-phase interaction coupling between the localized and propagating surface plasmons. That modification may help to tailor the absorption spectra and performance of the broadband MPA.

Figure 5 demonstrates the change in the relative absorption bandwidth and the average spectral absorption rate of the designed disc-in-hole MPA due to the modifications of nano-disc diameter d_1_ and nano-hole diameter d_2_.

The portion of Figure 5 below the white diagonal dashed line represents the situation where the titanium cap is entirely opaque, with no air holes. The vertical axis exemplifies the condition of the nano-hole cap array when d_1_ = 0. On that axis, as d_2_ increases, the structure exhibits a sharp absorption peak. In addition, the bandwidth becomes less expansive. It was expected that the extent of the opening enlarges the passage of light as a result of coupled propagating plasmonic modes on the slit surface. At the same time, the shrinkage of the lossy titanium minimizes the bandwidth. This result is consistent with that acquired in Figure 4b that when the nano-hole diameter d_2_ increases, the relative absorption bandwidth decreases. The results of Figure 5a exhibit the presence of region (I), the white dashed line, which is associated with the maximum available relative absorption bandwidth. That region, d_1_ ≈ 30 nm–40 nm and d_2_ ≈ 80 nm–90 nm, is designed to target the broad bandwidth. An average spectral absorption rate of 95% was achieved.

We numerically studied the macroscopic near-electric distributions to interpret the physical mechanism. Figure 6 presents the z-components of the electric field intensity, normalized to the maximum field intensity of the metamaterial absorber structure, with various cap arrays. One can see that both the localized plasmons on the nano-disc pattern, Figure 6a, and propagating surface plasmons on the inner surface of the nano-hole array, Figure 6b, can snare optical energy efficiently. Furthermore, the coupling between the localized and propagating plasmons are indicated in the proposed disc-in-hole arrays with three different dimensions in Figure 6c–e.

Short circuit current density, as one of the main parameters used to characterize the performance of solar cells, can be calculated using Equation (7) by assuming that an incident photon will produce an electron [9]:(7)$$J_{SC}=\frac{q}{hc}\overset{\lambda_{U}}{\underset{\lambda_{L}}{\int }}\lambda \;I\left(\lambda \right)\;A\left(\lambda \right)d\lambda$$
where *q* is the electron charge, *h* is the Planck constant, c is the speed of light, *I*(*λ*) is the incident light spectral power density, and *A*(*λ*) is the optical absorption. For standard solar cell measurements, the spectrum is standardized to the standard air mass 1.5 (AM 1.5 spectrum).

Finally, it is worth noting that the fabrication of nano-annular structures is a challenging process. However, several successful trials have been documented in the literature. The manufacturing process can be facilitated using electron-beam lithography, inductively coupled plasma deposition, metal sputtering, reactive ion etching, and chemical wet etching. The most important step in this fabrication process is to make the sidewalls of the ring-shaped pillars vertical to guarantee that no metals were deposited on the sidewalls during evaporation [34]. Table 1 compares the proposed plasmonic metamaterial absorber to structures documented in the literature, fabricated structures, and simulated work [26,36,38]. Regarding the reported absorber in this paper, it is very competitive due to its higher absorption and relative absorption bandwidth. Furthermore, its structural parameters can be changed independently, precisely, and easily.

It is worth noting that the wavelength-dependent angular response of MPA is essential to practical solar cell devices. This is due to a degradation in absorption levels as the incident angle increases. If the angular-dependent spectral responsivity is known, optical losses can be determined for any given angularly distributed solar spectrum.

## 4. Conclusions

In summary, we numerically studied the effect of embedding a disc-in-hole titanium–glass cap on the optical absorption spectra of a plasmonic metamaterial absorber. The impact of modifying nano-hole and embedded disc diameters on the absorber performance was investigated. The key argument is how the absorption spectra and performance of the broadband MPA could be tailored by tuning the out-of-phase interaction coupling between the localized and propagating surface plasmons. This can be done by modifying the inner nano-disc and outer nano-hole diameters. Our argument is supported by physical explanations and simulated local optical fields within the structure. The relative absorption bandwidth decreases as the nano-disc diameter increases. However, behaving in an antagonistic way, it increases as the nano-hole diameter increases. The proposed structure of the plasmonic metamaterial absorber reaches an average absorption of over 94% in a bandwidth of 0.81 µm and near-perfect absorption of 98% around the wavelength of 0.7 µm, with almost 100% relative absorption bandwidth and 41 mA/cm^2^ short circuit current density.

## Figures and Tables

**Figure 1 nanomaterials-12-03627-f001:**
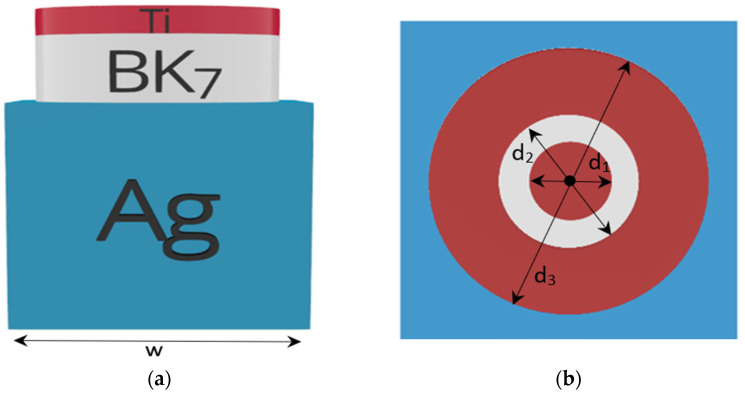
(**a**) Schematic diagram of a disc-in-hole PMA structure in 2D; (**b**) top view of the unit cell.

**Figure 2 nanomaterials-12-03627-f002:**
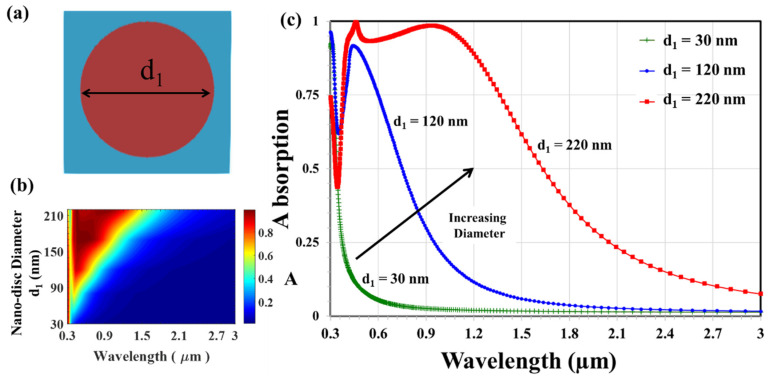
(**a**) Top view of nano-disc cap array and absorbance spectra (**b**,**c**) of the proposed plasmonic absorber.

**Figure 3 nanomaterials-12-03627-f003:**
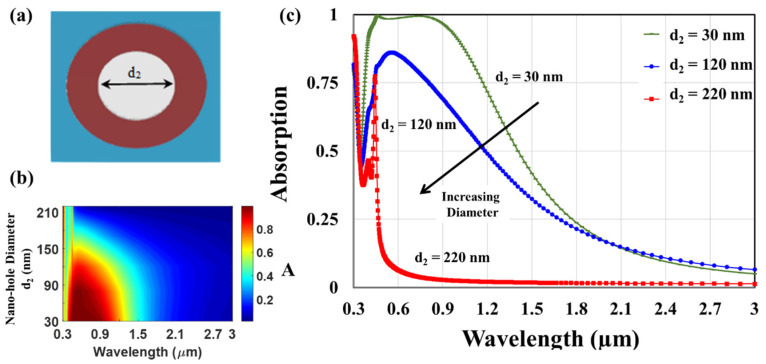
(**a**) Top view of nano-hole cap array and absorbance spectra (**b**,**c)** of the proposed plasmonic absorber.

**Figure 4 nanomaterials-12-03627-f004:**
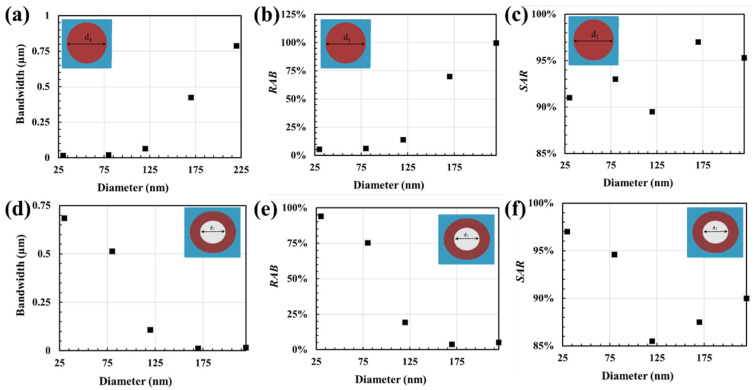
Absorption properties of the metamaterial absorber structure with various cap array (**a**,**d**) Bandwidth, (**b**,**e**) RAB, and (**c**,**f**) SAR.

**Figure 5 nanomaterials-12-03627-f005:**
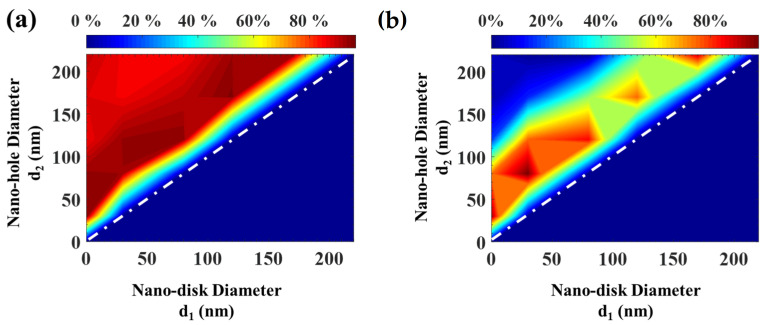
Modifications of the (**a**) RAB and (**b**) SAR of the designed disc-in-hole MPA due to changes in d_1_ and d_2_.

**Figure 6 nanomaterials-12-03627-f006:**
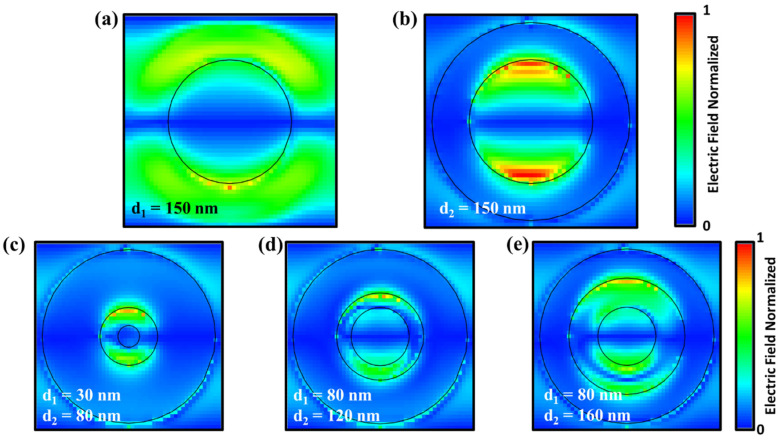
The spatial distribution, in the xy-plane at the middle of the metallic cap, of the z-component of electric field intensity, normalized to the maximum field intensity in MPA based on (**a**) nano-disc, (**b**) nano-hole, and (**c**–**e**) disc-in-hole arrays.

**Table 1 nanomaterials-12-03627-t001:** Absorption properties of the proposed absorber and some related works.

Structure and Dimension			Range, Bandwidth, and Relative Bandwidth with Absorption >85%				Maximum, Minimum, and Average Absorption within the Range			Short Circuit Current Density
Structure	d_1_ (nm)	d_2_ (nm)	*λ_L_* (µm)	*λ_U_* (µm)	*BW* (µm)	*RAB* (%)	*Amin*	*Amax*	*SAR*	*J_sc_* (mA/cm^2^)
disc -in- hole	30	80	0.42	1.23	0.81	98%	85%	98%	94%	41
	80	120	0.44	1.08	0.65	85%	85%	100%	95%	38
	120	170	0.45	0.98	0.53	75%	85%	100%	96%	36
[26] Ti–SiO_2_–Al (cubic cap)					0.71			99.8	97%	
[36] Axe-shaped resonator			0.32	0.98	0.66	100%		97%		
[38] Graphene–Si–back oxide (silicon nanowire)										38

## Data Availability

Not applicable.

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
