# Peer review of "Investigating the Absorption Spectra of a Plasmonic Metamaterial Absorber Based on Disc-in-Hole Nanometallic Structure"

_nanomaterials, 2022, doi:10.3390/nano12203627_

Round 1
Reviewer 1 Report
Comments to the author
In this manuscript, the authors numerically explore the absorption performance of a plasmonic metamaterial absorber. They use finite difference time domain (FDTD) to explore the impact on two of the design parameters of an absorber: diameter of the nano-hold and diameter of the embedded disc, in the optical and infrared range. By tuning these two parameters in their numerical model, they show that it is possible to design such an absorber with both high absorption and over a wide bandwidth, 94% over a bandwidth of 0.81 um. Overall, the systematic study of these two design parameters to drive design of high absorption plasmonic metamaterials is of interest to the community in the context of increasing absorption on solar cells through their incorporation. The work is well-presented and the numerical simulation framework and the accompanying analysis are clear. As such I would support publication of a revised version of this manuscript in Nanomaterials if the points below are properly addressed. I have listed below questions and comments that the authors should address before submitting the revision.
Major comments
1. [Page 4, line 117-120]: In referencing Figure 2 (c), an explanation of the crest is provided. However, the obvious dip for all three traces at wavelength = 0.36 um is not explained.
2. [Page 5, Figure 3 (c)] This is one of the key plots in this work. However, a lot of the features in the plots are left explained. For example, for the red trace (d2 = 120 nm), the absorption peak at 0.6 um and for the blue trace (d2 = 220 nm) the absorption peak at 0.5 um are not explained. More importantly, the pronounced dip at 0.36 - 0.45 um for the red trace (d2 = 120 nm) is unexplained as well.
3. [Page 5, Figure 4] How are the dashed lines in these plots obtained? Are they from an analytical model with fit parameters? If so, what are those parameters and what are the justification for their values?
4. [Page 6, Figure 4 (a), (c), (e)]: For these figures, which explore the dependence of the various absorption properties of the absorber on the nano-disc diameter, d1, what is value of the nano-hole diameter, d2, used here? Is it set to d2 = 0?
5. [Page 6, Figure 4 (b), (d), (f)]: Similarly, for these figures, which explore the dependence of the various absorption properties of the absorber on the nano-hole diameter, d2, what is value of the nano-hole diameter, d1, used here? Is it set to d1 = 0?
6. [Page 6, Figure 4 (e)]: Can the authors explain this strange dipping of the SAR at around 125 nm and then rising back up again?
7. [Page 6, Figure 4 (f)]: Similar to the point above, what explains this strange behavior of SAR dipping at 126 nm and then coming back up again?
8. [Page 6, line 156-159]:“Modifying the inner nano-disc diameter and outer nano-hole diameter may tune the out-of-phase interaction coupling between the localized and propagating surface plasmons. That modification may help to tailor the absorption spectra and performance of the broadband MPA.”
This is a key argument that is brought up several times in the paper. However, it is not supported with either an appropriate citation or simulation. For instance, a plot of the simulated local optical field within the structure would be extremely helpful in substantiating this claim.
9. [Page 6, Figure 5 (b)] The light green region around d1= 100 nm and d2 = 150 nm is puzzling and unexpected. Perhaps the authors are under sampling the parameter space in their simulation. Also, the dip and eventual rise of the SAB around either d1 = 125 nm or d2 = 125 nm in Fig. 4 (e) and (f), respectively, are not present in this plot where both d1 and d2 are varied. Why is that?
10. [Page 7, line 191-192]: “Table 1 compares the proposed plasmonic metamaterial absorber to structures in the literature, fabricated structures and simulated work.”
While I appreciate the author for providing such comparison, the paper could really benefit from a brief description of what the cited/ compared structures are.
Minor comments
1. [Page 4, Figure 2 (b)]: The color scale on this figure should be labeled.
2. [Page 5, Figure 3 (b)]: The color scale on this figure should be labeled.
3. [Page 6, Figure 4]: The ordering of the figure label of the is unconventional and should be fixed (Left to right and then top to bottom going from a-f).
4. [Page 6, Figure 4(a), (c), (e)]: The label d3 of the inset nano-disc image is hard to read. The font size needs to be increased. Also, to be consistent with the terminology used in the rest of the paper, it should read d1 instead of d3.

Author Response
We would like to express our sincere gratitude to the anonymous reviewer for his/her time and valuable comments, and to the Deputy Editor for handling our paper. We have revised the paper by seriously taking into account the comments provided. All the revisions in the revised manuscript are marked in blue color. Our point-to-point response is given below.

Reviewer 2 Report
Comments on the manuscript
In this manuscript entitled “Investigating the Absorption Spectra of a Plasmonic Metamaterial Absorber Based on Disc-in-Hole Nanometallic Structure,” the authors presented a systematic study on metal-insulator-metal metasurface absorbers. Specifically, the absorbers are composed of nanopatterned Ti-SiO2-metal ground plane configuration. The manuscript, in general, is interesting and their simulation results are incremental to the field of metasurface absorbers. Besides, the language is readable and fluent, and the references are appropriate. However, this manuscript did not provide deep insight into physics but instead presented superficial simulation results. This degrades the whole level of the manuscript.
Thus, this manuscript needs substantial revision before it can meet the scope of Nanomaterials. My comments and suggestion to the authors are listed below.
1. My major concern is related to the novelty of the implementation of an array of Ti rings or Ti circles configuration. Although this structure is relatively new, its performances, regarding the bandwidth and absorption coefficient, are neither innovative nor attractive. Similar MIM configurations have been extensively studied in experiments and theoretical explorations. For example, [Aydin, Koray, et al. "Broadband polarization-independent resonant light absorption using ultrathin plasmonic super absorbers." Nature communications 2.1 (2011): 1-7]. Thus, what is the key selling point for the new configuration in this work? Better absorption manipulation, wider bandwidth, more friendly fabrication, or others? The authors need to think seriously about these comments and questions, and briefly highlight the selling point in the abstract since it is closely related to the technical innovations and scientific impact of the manuscript.
2. “Recently plasmonic metamaterials have drawn attention due to it is unique absorption properties.” Need to include some works to support your claim. [Liang, Yao, et al. "Hybrid anisotropic plasmonic metasurfaces with multiple resonances of focused light beams." Nano Letters 21.20 (2021): 8917-8923].
3. Have the authors considered angular dispersion effects or angular stability in their design? Please comment.
4. It seems that the authors are not familiar with academic writing.
1) The introduction lists a lot of facts but no summarized key points. It is more like an undergraduate lab report than an academic abstract.
2) Introduction section.“ In this work, we present and explore the characteristics of a plasmonic metamaterial absorber based on a metal-insulator-metal (MIM) functional stack. The proposed structure consists of a BK7 glass “sandwiched” between a silver reflector and titanium metallic disc embedded inside a Ti periodic nano-hole array as a top layer. In the visible and infrared regimes, the optical absorption spectra of such structures have been investigated using the finite difference time domain (FDTD) method” First, regarding the BK7 glass, no one appreciates the reductant details of glass (BK7 or other kinds of glass) since this work does not contain any experimental study. Second, no need to give the abbreviations of MIM and FDTD, since they only appear once in the whole abstract.
5. “The plasmonic resonance causes an increase of efficiency in the visible region of the optical spectrum for Au and Ag, but in the ultraviolet region for the Al [22] and the plasmonic resonance enhances the efficiency in the visible and near-infrared for Ti [23].” What did you mean by using “an increase of efficiency”? According to my knowledge, absorption efficiency relies more on the configuration of nanostructures than materials. Please comment.
6. Figure 5. The data points seem ultra-rough.
Author Response

(The authors gave the same response as above.)

Round 2
Reviewer 2 Report
The authors have addressed my comments and improved the manuscript substantially. The manuscript is scientifically sound. I, therefore, have no more queries other than to offer my suggestion of acceptance. Good luck to the authors.